# Coupling Effect of Non-Ignition Impact and Heat on the Decay of FOX-7

**DOI:** 10.3390/molecules27238255

**Published:** 2022-11-26

**Authors:** Chongchong She, Kun Chen, Minglei Chen, Zhiyan Lu, Nana Wu, Lijie Li, Junfeng Wang, Shaohua Jin

**Affiliations:** 1School of Materials Science and Engineering, Beijing Institute of Technology, Beijing 100081, China; 2Gansu Yin Guang Chemical Industry Group Co., Ltd., Baiyin 730900, China

**Keywords:** FOX-7, thermal decomposition, ReaxFF, non-ignition impact

## Abstract

Non-ignition impact and heat stimuli are the most common external stimuli loaded on energetic materials. Nevertheless, there is thereby an urgent need, but it is still a significant challenge to comprehend their coupling effects on the decay and safety mechanisms of energetic materials. Then, reactive molecular dynamics simulation was employed to mimic practical situations and reveal the impact heat coupling effect on the decay mechanism of FOX-7. The temperature and the degree of compression of the crystal caused by the impact are considered variables in the simulation. Both increasing the degree of compression and elevating the temperature promotes the decay of FOX-7. However, their underlying response mechanism is not the same. The acceleration of decomposition is due to the elevated potential energy of the FOX-7 molecules because of elevating the temperature. In addition to the elevated potential energy of the molecule, the main contribution to the decomposition from the compression is to change the decomposition path. The results of the analysis show that compression reduces the stability of the C=C bond, so that chemical reactions related to the double bond occur. In addition, interestingly, the compression along the *c* direction has an almost equal effect on the final product as the compression along the *b* direction. Finally, the decay reaction networks are proposed to provide insights into the decomposition mechanism on atomic level. All these findings are expected to pave a way to understand the underlying response mechanism for the FOX-7 against external stimuli.

## 1. Introduction

Energetic materials are known for their rapid chemical decomposition along with a large amount of energy release, which may cause numerous unexpected accidents with more or less serious consequences. It is essential to detail the chemical reaction mechanism and to understand events that occur under external stimuli because of the extraordinary complexity of the initiation of detonation. Studies on the decomposition of energetic materials at the atomic level and interfacial structure variations were performed to understand the safety mechanism sufficiently [1,2,3,4,5]. These pieces of information can be used to establish the relationships between structures and performances, and then to evaluate the sensitivity to various external stimuli. Hence, the sensitivity of energetic materials should be treated as a long-term hottest topic due to it involving multiple factors.

1,1-Diamino-2,2-dinitroethylene, known as FOX-7 or DADNE, has attracted much attention owing to its outstanding properties of low sensitivity and high energy output. In general, the sensitivity response to external stimulation is usually used to evaluate the safety of energetic material. FOX-7 can be seen as a compromised energetic material of energy sensitivity contradictions with a similar detonation velocity to hexogeon (RDX) and a similar sensitivity to 1,3,5-triamino-2,4,6-trinitrobenzene (TATB) [6]. Additionally, this compromise allows FOX-7 to be widely used as an insensitive high-energy ingredient in PBX formulations. The FOX-7 unit cell consists of nearly planar molecules with a P2_1_/*n* space group [7,8]. The strong intramolecular H-bonding (HB) makes FOX-7 stable and also is the main reason for the low sensitivity of FOX-7. Additionally, the π-stacked interactions among the wavelike layer are very helpful to decrease the sensitivity and enhance the stabilities. For instance, the H_50,_ representing a drop high value with a probability of 50% explosion of explosives [9,10,11], of FOX-7 is as high as 89 cm, much higher than the 25 cm of RDX, suggesting a strong ability to resist external impact stimuli. Zhang et al. explored the influence of π-stacked interaction in FOX-7, TATB, and HMX on their abilities against external mechanical stimuli with molecular simulations [12]. Their work showed that π-stacked structures, particularly the planar layers, can effectively buffer against external mechanical stimuli by converting the mechanical energy acting on the explosive into intermolecular interaction energy. As a matter of fact, FOX-7 has already been regarded as one of the new generations of insensitive energetic materials [13].

To date, efforts have been devoted to exploring the responses and mechanisms of FOX-7 against heating and pressure [14,15,16,17,18,19,20,21]. With respect to heat-induced transformation, Bu et al. [22] explored the evolution of the structure in the polymorphic transformation from the α- to β- and γ-FOX-7. Their calculations suggested that the crystal packing varies from a wavelike shape to a face-to-face one as maximal torsion angles of O-N-C-C decrease from 35.6° to 25.6° and 20.2°, respectively. The primary thermal decomposition mechanism of FOX-7 was studied by Jiang et al. [23] with ReaxFF coupled with online photoionization mass spectrometry. The cleavage of C-NO_2_ was considered the initial decomposition step followed by the formation of C=O. Matthew et al. [24] determined the pressure–temperature phase diagram of FOX-7 by infrared radiation spectroscopy with the resistively heated diamond anvil cell (DAC) technique and characterized the decomposition products. For high pressure, Tao et al. [25] found that two phase transitions occurred in FOX-7 at 2 and 4.5 GPa and suggested that both transitions are attributed to the changes in hydrogen bonding. In other words, FOX-7 resists external stimuli by changing molecular arrangement. By the virtue of the density functional theory, Maija et al. [26] reported that shear plays a crucial role in the dissociation of molecules in organic energetic crystals under shear strain-induced effect.

Although the insensitive mechanism of Fox-7 has been demonstrated, the decomposition reaction has not been well understood. Moreover, multiple types of external stimuli are usually loaded on FOX-7 crystal simultaneously, and thus they have a coupled effect on its decomposition. Heat stimulation and low-velocity impact are the most common external stimuli during the transportation and storage of explosives. The low-velocity impacts tend to compress the explosive crystals, resulting in an increased probability of ignition. Relatively few studies on decay details of FOX-7, concerning the coupling effect of heat stimulation and low-velocity impact, have been published yet. ReaxFF molecular dynamics simulations (RMD) have been shown to reveal the decomposition mechanism of energetic materials [27,28,29]. Herein, the RMD is employed to investigate the effect of heat coupled with compression in different directions on the decomposition reaction network, the final products, and the reaction kinetic parameters of the FOX-7 decomposition. The calculated results are expected to shed some light on the chemical and mechanical properties when FOX-7 undergoes complex loadings.

## 2. Methodology

### 2.1. Reactive Force Field

The main difference between traditional unreactive force fields and ReaxFF is that the ReaxFF can provide accurate descriptions of bond breaking and formation as they occur during molecular dynamics (MD) simulations. ReaxFF uses a bond order–bond distance relation in conjunction with the bond order–bond energy relation [30,31], where contributions to the sigma, π, and double-π bonds are computed from the interatomic distances that are updated every MD step. Subsequently, the bond order is used to compute various partial energy contributions to the overall system energy, as given by Equation (1).
(1)$$E_{Reax}=E_{bond}+E_{lp}+E_{over}+E_{under}+E_{val}+E_{pen}+E_{tors}+E_{conj}+E_{H-bond}+E_{vdw}+E_{Coulomb}\;$$
where *E_bond_* is the bond energy; *E_lp_* is the lone pair energy; *E_over_* and *E_under_* are over- and under-coordination corrections, respectively; *E_val_* is the valence angle energy; *E_pen_* is the penalty energy; E_tors_ is the torsion conjugation energy; E_conj_ is the conjugation energy; *E_H-bond_* is the hydrogen bond energy; *E_vdw_* is the van der Waals energy, and *E_Coulomb_* is the electrostatic energy. However, the ReaxFF proposed by Duin [32] poorly describes the molecular crystal density. Liu [33] modified the parts of ReaxFF by improving the long-range dispersion to obtain the correct density for crystals (Equation (2)) based on the *E_Reax_*.
(2)$$E_{Reax-lg}=E_{Reax}+E_{lg}$$
where *E_lg_* is the long-range correction terms using the low-gradient model. After modification, the results calculated for RDX, PETN, TATB, and NM with ReaxFF-lg are in good agreement with experiment results. With respect to FOX-7 crystal, complete details of the verification of ReaxFF-lg will be provided in Section 3.1. The results show that the ReaxFF-lg may poorly predict FOX-7 crystallographic parameters after test. Herein, the ReaxFF-F parameters developed by James [34] through augmenting training set to include highly accurate intermolecular interaction energies are employed to simulate in our work. 

### 2.2. Hirshfeld Surface

Chemically, intermolecular interactions in explosives crystals may manifest as hydrogen bonding, π-stacking, and other weak interactions. The Hirshfeld surface method serves as a straightforward tool to visualize intermolecular interactions and crystal packing patterns instantly and to qualitatively and quantitatively encode information about the close contact between atoms. Surface shape depends on the interactions between molecules in the crystal as well as between atoms in molecules, and surface features reflect the interactions between different atomic sizes and intermolecular contact distances, as well as intermolecular interactions, in very subtle ways. The Hirshfeld surface of a molecule in a crystal is constructed by dividing the space in the crystal into regions in which the electron distribution of the spherical atomic sum of the pro-molecule dominates the corresponding sum on the original crystal. Following Hirshfeld, a molecular weight function (*w(r)*) was defined as [35]:(3)$$w\left(r\right)=\frac{\rho_{promolecule}}{\rho_{procrystal}}$$
(4)$$w\left(r\right)=\frac{\sum_{a\in molecule}\rho_{a}\left(r\right)}{\sum_{b\in crystal}\rho_{b}\left(r\right)}$$
where $\rho_{a}\left(r\right)$ is a spherically averaged atomic electron density centered on nucleus aaa, and the promolecule and procrystal are sums over the atoms belonging to the molecule and to the crystal, respectively. The Hirshfeld surface is then defined in a crystal as that region around a molecule where $w\left(r\right)\geq \frac{1}{2}$. That is, the region where the pro-molecule contribution to the pro-crystal electron density exceeds that of all other molecules in the crystal. Afterward, the distance from the surface to the nearest atom interior to the surface (*d_i_*), and the distance from the surface to the nearest atom exterior to the surface (*d_e_*) were defined to map Hirshfeld’s surface. Then, *d_i_* and *d_e_* were normalized by the van der Waals (vdW) radius (Equation (5)).
(5)$$d_{norm}=\frac{d_{i}-d_{i}^{vdw}}{r_{i}^{vdw}}+\frac{d_{e}-r_{e}^{vdw}}{r_{e}^{vdw}}$$
where $r_{i}^{vdw}$ and $r_{e}^{vdw}\;$ represent the vdW radii of the atoms closest to the points inside and outside the surface, respectively. According to *d_norm_*, the regions with larger contributions to intermolecular interactions can be identified. Mapping these (*d_i_*, *d_e_*) points on Hirshfeld surface and considering their relative frequencies, a 2D fingerprint plot can be obtained. Based on the 2D fingerprint plot, the composition of intermolecular interactions can be clearly ascertained. All the surfaces and fingerprint plots were created using CrystalExplorer package [36].

### 2.3. Compute Details

The original unit cell was obtained directly from the Cambridge structure database (CSD ref code: SEDTUQ) and then was expanded into periodic supercell with 300 FOX-7 molecules. The schematic diagram of the simulation process is shown in Figure 1. Firstly, a conjugate gradient algorithm and a 10 ps canonical ensemble (NVT) MD simulation at 300 K were used to relax the internal stress of the FOX-7 supercells [23,27,37,38]. The relaxed supercell was compressed and performed with isothermal–isobaric (NPT) MD simulation, and the equilibrium structure was obtained. In order to mimic the non-ignition impact, 4 NPT at 300 K with different pressure were carried to compress the FOX-7 supercell. The 4 compressed supercells are denoted as layer0.8, layer0.9, s0.8, and s0.9, respectively, in which the “layer” represents the compression along the c direction with “a” and “b” fixed; the “s” represents the compression along the b direction with “a” and “b” fixed; “0.8” means the compressed supercell is 0.8 V_0_. Then, the 5 supercells are used as the initial structure for subsequent MD simulations, which include 5 ps NVT MD simulations at 300 K to equilibrate and 200 ps NVT MD simulations at 2000 K, 2500 K, 2750 K, 3000 K, and 3500 K. All MD simulations were performed using large-scale atomic/molecular massively parallel simulator (LAMMPS) program package. The Nose–Hoover thermostat is used in all NVT simulations. The atomic coordinates and bond information in simulation are recorded every 50 fs. ReacNetGenerator, an automatic reaction network generator developed by Zeng et al., is employed to extract the reaction network from the bonds files [39]. 

## 3. Results and Discussion

### 3.1. Validation of ReaxFF

Our previous work has verified that the ReaxFF-F accurately predicted the bond length in FOX-7 [40]. To verify the accuracy of ReaxFF parameters, NPT (1 atm) simulations with ReaxFF-lg and ReaxFF-F were carried out to simulate the FOX-7 crystal density, respectively. Then, the results were compared with the experiment data. The time-dependent curves of the density of FOX-7 crystal are exhibited in Figure 2a. For ReaxFF-lg, the density of the system decreases sharply from 1.92 g/cm^3^ to 1.23 g/cm^3^ at the initial stage of simulation and gradually increased, reaching equilibrium (1.51 g/cm^3^) with a 21% error compared to 1.907 g/cm^3^ form the experimental test. Meanwhile, the density of the system reached equilibrium (1.91 g/cm^3^) although the density also decreased slightly at the beginning of the simulations with ReaxFF-F. Obviously, the density of FOX-7 predicted by ReaxFF-F is consistent with the experimental results. The molecular arrangement in the simulations was also discussed. Although the molecular arrangement simulated by ReaxFF-F became disordered, the crystals reverted to the original layered 2D structure as the simulation progressed. In contrast, the molecules cannot revert to a regular arrangement in the simulations with ReaxFF-lg. As aforementioned, numerous hydrogen bonds exist in FOX-7 crystals. Therefore, this should be attributed to the fact that the ReaxFF-lg cannot describe the strong interactions among FOX-7 molecules in the crystal, although a dispersion correction was given in ReaxFF-lg in order to describe the structure of energetic materials such as RDX and HMX [41]. Different from ReacFF-lg, the parametrization of the van der Waals terms and EEM terms in ReaxFF-F was developed by augmenting the training set including highly accurate intermolecular interaction [42]. Additionally, both ReaxFF-lg and ReaxFF-F were used to calculate the energy of some molecules. It can be seen that the description of molecular energy by the two force fields is basically the same (Figure 2b). As a result, the ReaxFF-F is more suitable than ReaxFF-lg to investigate the decomposition of FOX-7.

### 3.2. Intermolecular Interactions

In fact, the initial decomposition of condensed-phase explosives is considered to be closely related to the formation of hot spots. The heat resulting from the local plastic deformation of explosives under impact loading is one of the mechanisms of hot spot generation. Moreover, the deformation of explosives is directly related to the slip motion among molecules. Due to the intermolecular interactions in the crystal, the slip motion will lead to a certain degree of deformation of the molecules, so that the molecules are excited to a high-energy vibrational state, which in turn triggers the breaking of chemical bonds. Therefore, understanding the intermolecular interactions under impact loading is the basis for exploring the decomposition mechanism of explosives.

Previous studies have shown that hydrogen bonding and π-stacking are beneficial to low-impact sensitivity [12,26,43]. The details of structures in five cases were illustrated in Figure 3. The interlayer spacing and the angle α relate to the interlayer molecules change significantly when the crystal is compressed along the c direction. The value of α increases from the original 137.03°@FOX-7 to 139.16°@Layer0.9 and then to 143.02°@Layer0.8. Meanwhile, the interlayer spacing (d) decreases from 3.07 Å to 2.80 Å and 2.44 Å. The deformation along the c direction mainly affects the interlayer spacing and the α, causing an obvious hydrogen bonding between the interlayer molecules in layer0.8. The hydrogen bonding between these interlayer molecules obviously increases the energy barrier for layer slip. In fact, the interlayered distance governs the sliding barrier and further the sensitivity. Thus, the crystals are more prone to decomposition as the interlayer spacing decreases. With regard to the crystal under the external force along the b direction, the dominance change is the value of α. A sharp decrease happened to α from 137.03° to 130.62°@s0.9, and then to 128.76°@s0.8, which can reduce the thermal stability of the molecule. Both the deformation along the c direction and the b direction have different degrees of damage to the FOX-7 molecules to a certain extent. However, the deformation along the c direction can be attenuated by the interaction between the layers.

To understand the effect of deformation on crystals clearly, the Hirshfeld surface and the related 2D fingerprint plots were employed to visualize the intermolecular interactions while keeping the FOX-7 molecular orientations as consistent as possible (Figure 4). In general, the distribution of red dots on the Hirshfeld surface shape is dependent on the distance of close contacts and representing the environment of a molecule in a crystal. The molecular positions corresponding to these red regions may be where hot spots are generated and further decomposition occurs. Additionally, the blue area on the surface represents a considerable intermolecular distance, beneficial to ready shear sliding and further low-impact sensitivity [44]. As the degree of compression along the c direction increases, remarkably, red dots appear on the Hirshfeld surface (Figure 4b,c). Moreover, the appearing red region mainly corresponds to the nitro group and the carbon atom, suggesting that these locations may be the initial locations where decomposition occurs. For the compression along the b direction, it can be clearly seen that no red area appears on the upper surface of the molecule, but the red area on the side of the Hirshfeld surface increases dramatically (Figure 4d,e). In other words, the distance between the nitro group and the amine group in the layer decreases, which implies that the probability of H transfer to NO_2_ increases. Screening the composition of intermolecular interactions shown from 2D fingerprint (Figure 4a–e), the H···O interaction was always predominant in all systems. However, when subjected to compression in different directions, the composition of intermolecular interactions changes differently (Figure 4f). It can be seen that along the c direction, the proportion of O···H hydrogen bonding interactions is relatively reduced, and the N···H and other interactions including C···H and C···O increase. Whereas along the b direction, the N-H interaction decreases, the N···O interaction increases, and the other interactions are almost unchanged. 

### 3.3. Evolution of Chemical Species

As to the decay details of FOX-7, the evolution of the main related chemical species was also explored, including FOX-7, NO_2_, NH_3_, CO_2_, H_2_O, N_2_, and H_2_. First, the evolution of FOX-7 under various conditions was concerned to investigate the effect of temperature and compression on the decomposition of FOX-7 in Figure 5. As a whole, FOX-7 crystal is fully decayed at around 20~30 ps under 2000 K, whereas it is rapidly decomposed within 2 ps under 3500 K. It can be concluded readily that higher temperature leads to faster decomposition of FOX-7. However, the effect of compression on decomposition is quite different from the temperature. The compression along the layer stacking direction (c direction), the decomposition rate in layer0.8@2000 K and layer0.9@2000 K is generally slower except for the initial decomposition stage relative to FOX-7@2000 K, implying that the greater degree of compression results in slower decomposition rate. In contrast, the compression along the b direction accelerates the decomposition of FOX-7 crystals. For the reason resulting in the slowed-down decomposition of layer0.8@2000 K and layer0.9@2000 K, we speculate that the compression makes the FOX-7 molecules tend to have a planar distribution similar to TATB so that the thermal stimulation effect reduces the contribution of molecular twist, which is the main trigger for the decomposition. In addition, it is interesting that this phenomenon disappears at high temperatures. For example, at 3000 K, no matter which direction the FOX-7 crystal is compressed from, the compression facilitates the decay of FOX-7. The order of decomposition rate is FOX-7, layer0.9, layer0.8, s0.9, and s0.8, implying that higher compression causes a faster decomposition rate. Note that this does not imply that the compression of FOX-7 along the c direction is beneficial for the thermal stability of FOX-7. It can be clearly seen that compression in the initial stage of decomposition has a promoting effect on decomposition, which may be responsible for the increased probability of ignition for FOX-7 crystal under thermal stimulation after suffering an external non-ignition impact.

Furthermore, the reaction kinetic parameters were calculated to evaluate the effects of temperature and compression on the decomposition of the FOX-7 crystals [37]. The number of FOX-7 in the initial decomposition reaction can be described by the first-order decay exponent *N(t)* (Equation (6)).
(6)$$N\left(t\right)=N_{0}\left(1-\exp\left(-k_{1}t\right)\right)$$
where *t*_0_ are the times at which FOX-7 begins to decompose; *k*_1_ is the reaction rate; *N*_0_ and *N_t_* is the initial number of FOX-7 at *t*_0_ and *t*. Then, the kinetics parameters can be employed to calculate the activation barrier according to the Arrhenius law (Equation (7)).
(7)$$lnk=lnA- [\frac{E_{a}}{R}]\frac{1}{T}\;$$
where *A* is the exponential pre-factor, *E_a_* is the activation barrier. The *lnk~1/T* was obtained by fitting the *k*_1_. The linear corrections of *lnk*_1_ and *1/T* were shown in Figure 6a. It can be clearly seen that the $\frac{E_{a}}{R}$ of the crystal decreases slightly when the FOX-7 crystal is compressed along the c direction, while the compression along the b direction the $\frac{E_{a}}{R}$ decreases from 18.2@FOX-7 to 3.89@s0.8 and 9.32@s0.9, respectively. In other words, compression along the b direction will make FOX-7 crystals possess lower activation energy. However, $\frac{E_{a}}{R}$ increase slightly in layer0.9 and layer0.8, which is unexpected and contrary to our previous speculation. In order to explain this phenomenon, the correlation between the molecular angle in the layer (α) and $\frac{E_{a}}{R}$ was fitted. It can be seen that the value of α is positively correlated with $\frac{E_{a}}{R}$ (Figure 6b). To a certain extent, this shows that the planar structure is more resistant to shock compression than the wave structure. It should be pointed out that the discussion in this paper is about the non-ignition impact on the FOX-7 crystal. That is, the impact only compresses the crystal and does not initiate ignition. The relationship between structure and the value of $\frac{E_{a}}{R}$ may help us understand the microscopic ignition mechanism of FOX-7 crystals under non-ignition impact (low-velocity impact) and give us new ideas to design new low-sensitivity and high-energy energetic materials, such as changing the molecular arrangement in the crystal to be planar by eutectic technology. 

There are three major steps for igniting the FOX-7, including the hydrogen transfer from NH_2_ to NO_2_ along with the formation of HONO, and the C-N bond fission to form NO_2_ and NH_2_. Among these steps, the NO_2_ partition and the hydrogen transfer dominate the initial decay steps in FOX-7 crystal. It is well-known that the dissociation of C-NO_2_ is an important part of the initial decomposition reaction in the energetic materials. Previous studies have shown that temperature can accelerate the dissociation of C-NO_2_ and shorten the lifetime of NO_2_ [42,45,46]. Therefore, NO_2_ increases first and then decreases in the initial decomposition stage, which is consistent with the present results. Under 2000 K, n(NO_2_) even fluctuates around 50 ps for a long time, whereas NO_2_ is present for only ~15 ps at 3500 K; for the other cases, the lifetime of NO_2_ is between the two cases. Then, the effect of compression on the dissociation reaction of NO_2_ is mainly concerned. From the evolution of NO_2_ in Appendix A, it is interesting to find that the compression on the FOX-7 crystal tends to inhibit the dissociation of C-NO_2_. For example, the maximum n(NO_2_) decreased from 48@FOX-7 to 35@layer0.9, 26@s0.9, 22@layer0.8, and 17@s0.8 at 3500 K. Two reasons may be responsible for the reduction of NO_2_. On the one hand, the compression on FOX-7 crystals results in a small intermolecular distance, which may facilitate the nitro–nitrite (NO_2_→ONO) isomerization followed by the NO fission. On the other hand, the compression may lead to new decomposition paths in the decomposition of FOX-7, which will be further discussed below. In a summary, the presence of compression on the crystal may change the weight of related initial reaction steps. 

The number of final products can also reflect the effect of temperature compression on the decomposition process, which can reflect the degree of energy release of the energetic material to a certain extent. Appendix A shows the evolution of CO_2_, N_2_, H_2,_ and H_2_O in the five cases under various temperatures. The temperature has relatively little effect on the number of final products at a given compression, whereas compression distinguishes the evolution curves at a given temperature. In addition, the evolution curves of these final products in layer0.8 and s0.8 overlap each other. The number of CO_2_ in FOX-7, layer0.9 (s0.9), and layer0.8 (s0.8) is 398, 345, and 252, respectively, suggesting that compression along the c direction and b direction possess almost an equivalent effect on the number of final products. This trend also appears for other products of N_2_, H_2_, and H_2_O but not in NH_3_. As for H_2_O, n(H_2_O) increase rapidly to maximum firstly and decrease gradually to equilibrium till the end of the simulation, which is caused by its decomposition at high temperature. 

### 3.4. Details of Decomposition Reaction Network

Details of the decomposition reaction network may be instructive to understand the detonation properties of FOX-7 under extreme conditions. The decomposition reaction is quite complex with thousands of species so it is unrealistic to manually analyze the reaction network. In fact, the main reaction mechanisms are often the focus. In ReacNetGenerator, the hidden Markov model (HMM) is used to remove the rare reaction and the short lifetime species in the decomposition process, making the analysis process easier and more accurate. The HMM parameters matrix A and B is tested and verified manually. When $A=\left(\begin{array}{cc} 0.6 & 0.4\\ 0.4 & 0.6 \end{array}\right)$, B and C are default values, the complexity of the reaction can be reduced, and a relatively accurate reaction network can be obtained. Figure 7 shows the reaction network analysis with no HMM and HMM based on the initial decomposition of FOX-7 under 2000 K. As can be seen, many similar structures with a very short lifetime appear in the analysis result with no HMM, for instance, structure 2, 5, 6, 11, and 12, which is caused by large-amplitude molecular vibrations and collisions. More importantly, these structures are not conducive to our analysis to obtain effective and accurate reaction information. After the HMM process, a relatively clear reaction network is shown in Figure 7b, which is in accord with the previous studies.

Subsequently, the effect of temperature on the decomposition path was analyzed. The initial reaction nets under five temperatures are shown in Appendix A. At 2000 K, based on the main decomposition products, such as molecules 6, 7, and 8 in Appendix A, it is suggested that the initial decomposition reaction is mainly intramolecular and intermolecular H transfer due to the smaller energy barrier required for H transfer. The C-N bond cleavage reactions with a high energy barrier are less frequent. However, the FOX-7 molecule framework is broken after the H transfer so that the energy for C-N dissociation decreases. Then, the FOX-7 structures that lose one H atom decompose through C-N dissociation, which is consistent with the slow rise of the NO_2_ curve under 2000 K in Appendix A. As the temperature was elevated, the proportion of C-NO_2_ bond cleavage in the initial reaction increased, which can be demonstrated by the increase in the number of HNO, HNO_2_, and NO_2_ [47]. In other words, the high temperature favors the dissociation of C-N bonds.

After analyzing the initial decomposition paths at different degrees of compression under 2000 K, it was found that the compression along the c direction changes the initial decomposition pathway by increasing the proportion of double bonds involved in the reaction except for facilitating H transfer, implying that the compression destabilizes the double bond. As can be seen from Figure 8a, for s0.8, the left side of the figure is the decomposition path of H transfer and C-N_2_ bond cleavage and the right side is the reaction involving the double bond. In addition to the increase in double bond-related reactions, the frequency of N-O dissociation is also significantly increased for s0.8. The twisting of the NO_2_ in the FOX-7 molecule may be responsible for the N-O dissociation. As a whole, the proportion of double bond cleavage in the initial decomposition reaction increases under compression–thermal stimulation resulting in different evolution of the final products. Finally, simplified decomposition networks at different compression levels in simulation time are shown in Figure 8b.

## 4. Conclusions

In summary, ReaxFF MD simulation was performed successfully to mimic non-ignition impact compression and heat stimulation loading on FOX-7 crystal. The simulation suggests that both the presence of compression and increasing the temperature promote the decay of FOX-7. Structural analysis of the compressed crystals shows that compression distorts the FOX-7 molecule, and the location of the twist may be the initial location where decomposition begins. More importantly, the crystals under the compression along the b direction have a faster decomposition rate than those compressed along the c direction. Even so, the compression at a given compression degree along the c direction and b direction possess an equivalent effect on the FOX-7 decay in terms of the evolutions of the number of the final products. Finally, the coupling effect of compress and heat stimulation on the initial decay path is proposed. The presence of compression may influence the stability of C=C bonds to accelerate FOX-7 decay. All these findings are expected to reveal the physical origin of the sensitivity enhancement under multiple stimuli for FOX-7. 

## Figures and Tables

**Figure 1 molecules-27-08255-f001:**
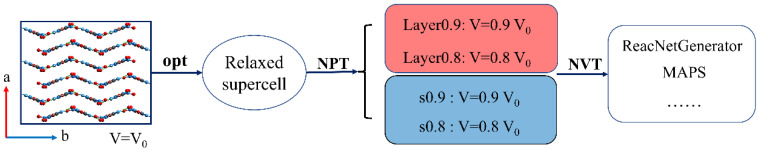
The schematic diagram of the simulation process. The red and blue color means the compression along c direction and b direction, respectively.

**Figure 2 molecules-27-08255-f002:**
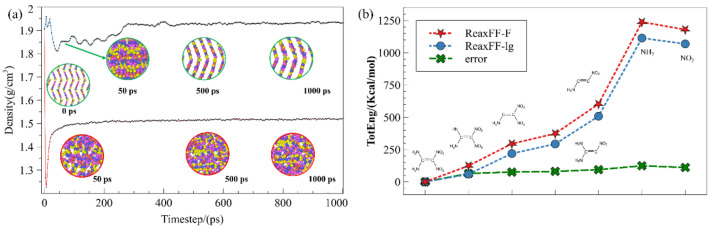
(**a**) the prediction of FOX-7 crystal density by two ReaxFF-lg and ReaxFF-F; (**b**) the description of molecular energy obtained from the two force fields.

**Figure 3 molecules-27-08255-f003:**
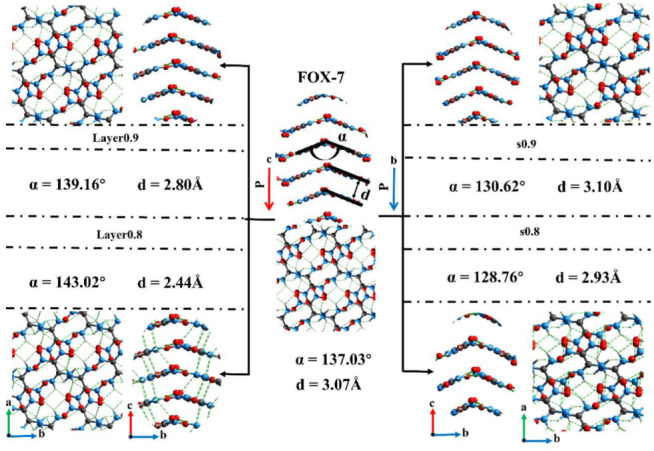
The details of structures in five cases. The left and right cases are compression along the c and b directions, respectively. The green dotted lines represent intermolecular hydrogen bonds.

**Figure 4 molecules-27-08255-f004:**
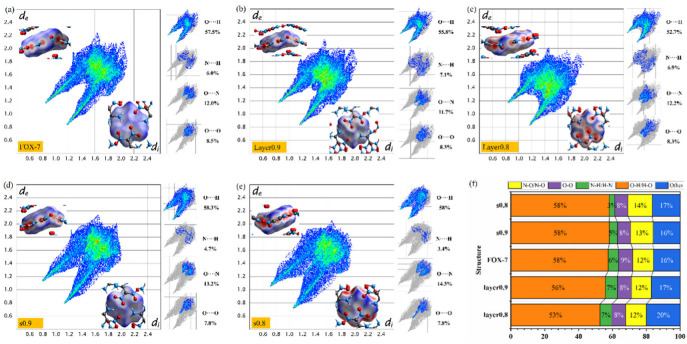
(**a**–**e**) The Hirshfeld surface and the related 2D fingerprint plots of five cases: FOX-7, Layer0.9, Layer0.8, s0.9, s0.8; (**f**) The composition of intermolecular interactions in the five case.

**Figure 5 molecules-27-08255-f005:**
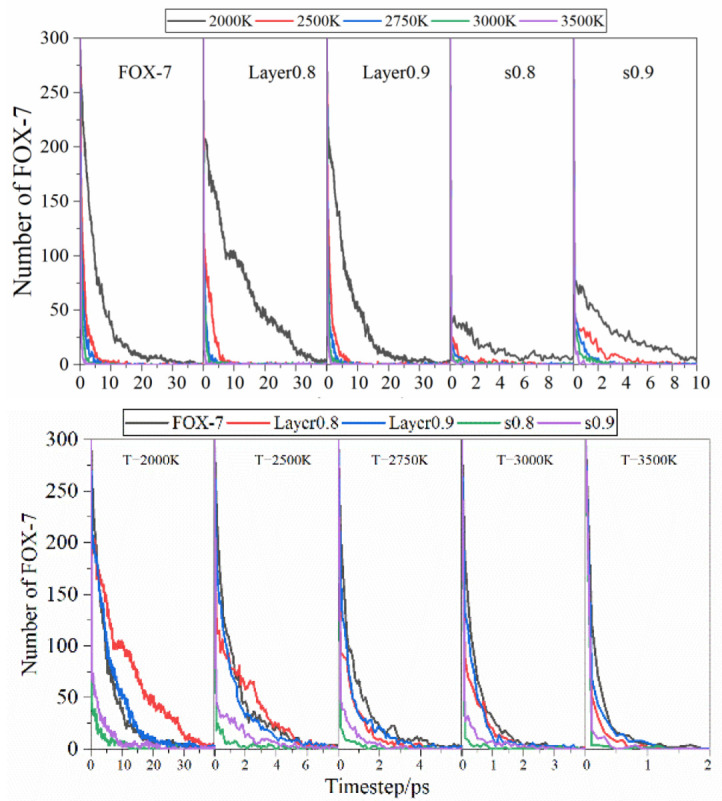
Evolution of the number of the FOX-7 molecules under various conditions.

**Figure 6 molecules-27-08255-f006:**
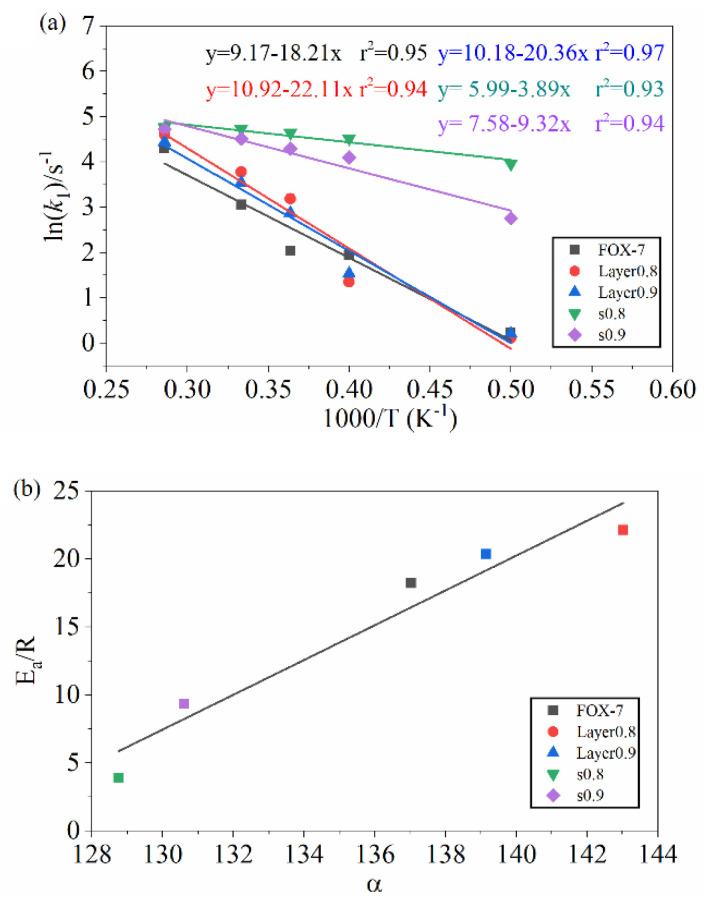
(**a**) Logarithm of FOX−7 decomposition rate constant *ln(k)* against *(1/T)* in range of 2000–3500 K. The solid point and solid line represent calculated and linearly fitted values. (**b**) The correlation between the (α) and $\frac{E_{a}}{R}$.

**Figure 7 molecules-27-08255-f007:**
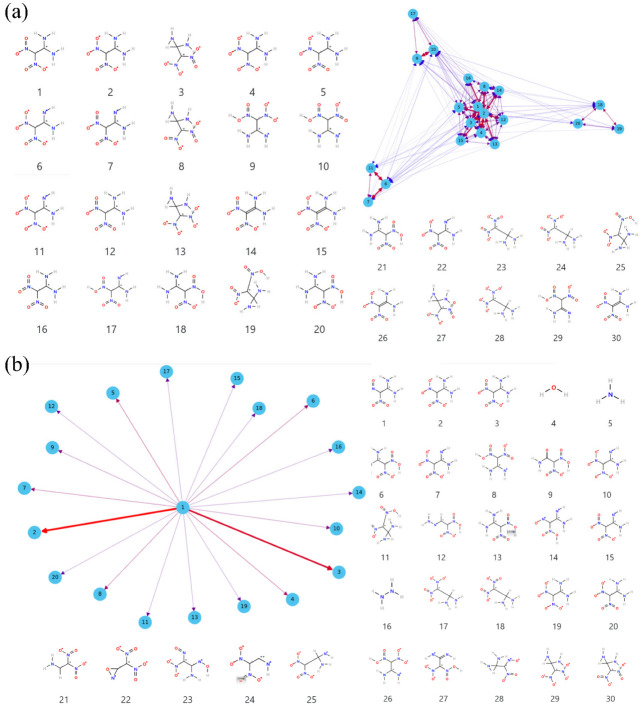
The initial reaction network for FOX-7 at 2000 K: (**a**) the network without HMM filtering; (**b**) the network with HMM.

**Figure 8 molecules-27-08255-f008:**
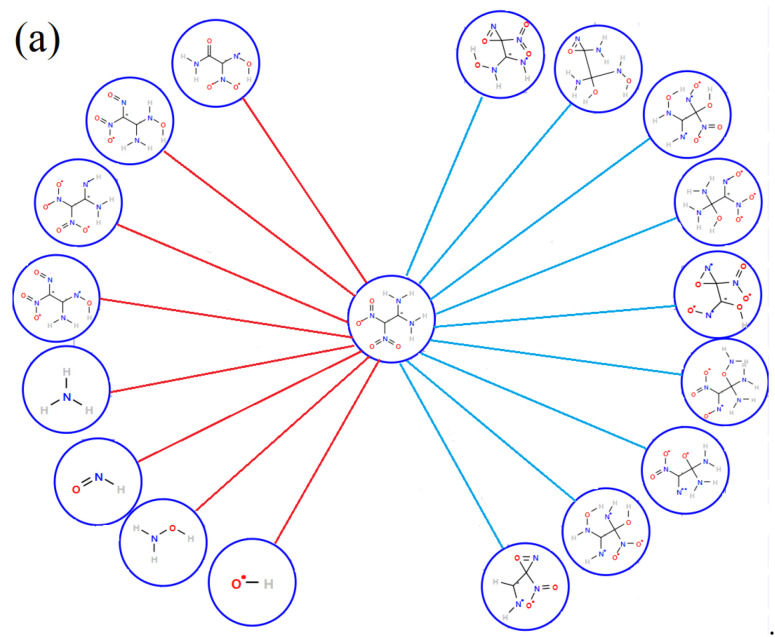
(**a**) The initial decomposition path in s0.8 under 2000 K. (**b**) The reaction network from FOX-7 to H_2_O, CO_2_, N_2_ … formed by the top 20 species.

## Data Availability

Not applicable.

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
