# Peer review of "Coupling Effect of Non-Ignition Impact and Heat on the Decay of FOX-7"

_molecules, 2022, doi:10.3390/molecules27238255_

Round 1
Reviewer 1 Report
In present paper, the coupling effects of non-ignition impact and heat on FOX-7 decay and their microscopic mechanisms were studied by reactive molecular dynamics simulation. It was found that increasing the compression degree and temperature would promote the decay of FOX-7, but their reaction mechanism was different. The acceleration of decomposition is caused by the increase of the potential energy of FOX-7 molecule due to the increase of temperature. Compression reduces the stability of the C=C bond, which leads to a chemical reaction related to the double bond. Compression in different directions has basically the same effect on the final product. The author gives a decomposition network. The research work can help for further understanding the response mechanism of FOX-7 to external stimuli. It is of great significance. It is very interesting. The overall design of the paper is detailed, the data is full and accurate, and the analysis is reasonable. However, the following suggestions need further consideration by the author, which will help to further improve the quality of the paper.
1. The authors should carefully check the entire paper, including the expression of formulas, the expression of variables in formulas, and the consistency. For example, what is k1?
2. The paper mainly reported the simulation results, but how can the authors ensure the reliability of the simulation parameters used in simulation and the results obtained. It is suggested to add some experimental data such as density and other properties to compare with the simulation results.
3. The resolution of Figures 7 and 8 is not enough, and the interpretation and analysis of these two figures are also not enough. It is recommended to modify them.
4. "Declaration of competing interest" and "Acknowledgements" should be regarded as “SECTION” of the paper. Moreover, the information (such as the grant number) for supporting grants should be improved.
5. References should be carefully checked to ensure accuracy and compliance with specifications.
Author Response
Dear Professor:
Thank you very much for giving us the opportunity to revise our manuscript. We also thank the reviewers for their valuable and helpful comments. Accordingly, we’ve addressed all comments – both specific and general – from two three reviewers. We have carefully taken their comments into consideration in preparing our revision, resulting in a clearer, more compelling, and broader expression. The following summarizes how we responded to the reviewers’ comments.
Response to Reviewer #1:
- 1. The authors should carefully check the entire paper, including the expression of formulas, the expression of variables in formulas, and the consistency. For example, what is k1?
Answer:
We check the entire paper carefully and correct the expression of formulas. All the modifications are highlight with red.
- The paper mainly reported the simulation results, but how can the authors ensure the reliability of the simulation parameters used in simulation and the results obtained. It is suggested to add some experimental data such as density and other properties to compare with the simulation results.
Answer:
The reaction force field has been widely used to simulate and study the decomposition process of energetic materials in various extreme environments such as high temperature and shock wave, and has given relatively reliable results. The crystal data in this paper are derived from experimental data in the crystal database, and the simulation parameter is based on the literatures. Experimental data such as density were added to the article.
- The resolution of Figures 7 and 8 is not enough, and the interpretation and analysis of these two figures are also not enough. It is recommended to modify them.
Answer:
Due to the larger image resulting from the large number of elements in the Figures 7 and 8, we slightly increased the resolution of Figure, and we can provide a higher resolution image if needed.
- Declaration of competing interest" and "Acknowledgements" should be regarded as “SECTION” of the paper. Moreover, the information (such as the grant number) for supporting grants should be improved.
Answer:
Declaration of competing interest" and "Acknowledgements" are added in the paper.
- References should be carefully checked to ensure accuracy and compliance with specifications.
Answer:
The References are checked and modified in the paper.
We marked all the revised sentences and phases in red for clear identifications. Uploaded is our revised manuscript. And we are willing to make any further revisions if needed.
Again, thank you very much for your consideration.
Sincerely yours,
Prof. Kun Chen

Reviewer 2 Report
Review of the ms "Coupling Effect of Non-Ignition Impact and Heat on the Decay of FOX-7" by Chongchong She et al.
The ms presents a theoretical study aiming at evaluating the role of heat and compression in the decay of the FOX-7, a very energetic material. The manuscript presents some interesting results, particularly for the practical applications of the title material. The work is essentially well-written and methodologically consistent, though some issues can be improved before accepting for publication.
Major issues
Result and Discussion
-Validation of ReaxFF.
The discussion related to the density is quite confusing, in turn, it appears to be an important validation step of the methodology followed.
Line 178. Which is the origin of the 21% error? Notice with such a value, the initial stage, and the equilibrium densities are inside the error interval.
Line 180, Why is so obvious the ReaxFF predicted density is consistent with experimental results? Which experimental values?
Lines 181-182. It seems the correct meaning of that sentence should be "The molecular arrangement in the simulations will be discussed in the following"
-Details of the decomposition reaction network
I suggest reporting the corresponding contribution (in percentiles, for example) of the more significant reaction decay pathways, at least for some selected temperatures.
Minor issues
Introduction
Line 50. The explosive sensitivity H50 should be properly introduced and referenced.
Methodology
Line 151. On which criteria was the relaxation time of 10ps selected?
Correct Hirsheld in line 232
Eqn 6. closing parenthesis missing
Author Response
Dear Professor:
Thank you very much for giving us the opportunity to revise our manuscript. We also thank the reviewers for their valuable and helpful comments. Accordingly, we’ve addressed all comments – both specific and general – from two three reviewers. We have carefully taken their comments into consideration in preparing our revision, resulting in a clearer, more compelling, and broader expression. The following summarizes how we responded to the reviewers’ comments.
Response to Reviewer #2:
- -Validation of ReaxFF.
The discussion related to the density is quite confusing, in turn, it appears to be an important validation step of the methodology followed.
Line 178. Which is the origin of the 21% error? Notice with such a value, the initial stage, and the equilibrium densities are inside the error interval.
Line 180, Why is so obvious the ReaxFF predicted density is consistent with experimental results? Which experimental values?
Lines 181-182. It seems the correct meaning of that sentence should be "The molecular arrangement in the simulations will be discussed in the following"
Answer:
(1) The 21% error is the density of the crystal after equilibrium simulation relative to the density obtained from the experimental test.
(2) The parametrization of the van der Waals terms and EEM terms in ReaxFF-F was developed by augmenting the training set including highly accurate intermolecular interaction, which can describe the hydrogen bond interaction in FOX-7 crystals better. The experimental values are added in the Methods section of the article.
(3) The discussion related to the density is modified and highlight in the paper.
- -Details of the decomposition reaction network
I suggest reporting the corresponding contribution (in percentiles, for example) of the more significant reaction decay pathways, at least for some selected temperatures.
Answer:
The analysis results in this paper are obtained by the ReacNetGenerator program, which cannot give the corresponding contribution of the more significant reaction decay pathways, which is also the content of our subsequent development research.
- Introduction
Line 50. The explosive sensitivity H50 should be properly introduced and referenced.
Answer:
An explanation of H50 has been added to the article and relevant references have been attached.
- Methodology
Line 151. On which criteria was the relaxation time of 10ps selected?
Answer:
The setting of the relaxation time selected is a reference to the literature and relevant literature is added to the article.
- Correct Hirsheld in line 232 Hirshfeld
Eqn 6. closing parenthesis missing
Answer:
We check the entire paper carefully and correct the mistakes. All the modifications are highlight with red.
We marked all the revised sentences and phases in red for clear identifications. Uploaded is our revised manuscript. And we are willing to make any further revisions if needed.
Again, thank you very much for your consideration.
Sincerely yours,
Prof. Kun Chen

Round 2
Reviewer 1 Report
The authors have reviesed the manuscript as all my comments. It can be accepted for publication after checking the text.eg. Ref. 1 "1. 1. Junying Wu", Ref. 6 to 9 etc. Again, it would be better to provide a higher resolution images for Fig.7 and Fig. 8.
Author Response
Dear Professor:
Thank you very much for giving us the opportunity to revise our manuscript. We also thank the reviewers for their valuable and helpful comments. Accordingly, we’ve addressed all comments – both specific and general – from two three reviewers. We have carefully taken their comments into consideration in preparing our revision, resulting in a clearer, more compelling, and broader expression. The following summarizes how we responded to the reviewers’ comments.
Response to Reviewer #1:
- The authors have reviesed the manuscript as all my comments. It can be accepted for publication after checking the text.eg. Ref. 1 "1. 1. Junying Wu", Ref. 6 to 9 etc. Again, it would be better to provide a higher resolution images for Fig.7 and Fig. 8.
Answer:
A figure with 800 dpi for Fig.7 and Fig. 8 is added in the paper and we can provide a higher resolution image if needed.
The text of all references was checked.
